# Estimation of Fusarium Head Blight Severity Based on Transfer Learning

Chunfeng Gao [1], Zheng Gong [1], Xingjie Ji [2,3], Mengjia Dang [1], Qiang He [1], Heguang Sun [1] and Wei Guo [1,*]

1    College of Information and Management Science, Henan Agricultural University, Zhengzhou 450046, China
2    Henan Key Laboratory of Agrometeorological Support and Applied Technique, Zhengzhou 450003, China
3    Henan Institute of Meteorological Sciences, Zhengzhou 450003, China
*    Correspondence: guowei@henau.edu.cn

**Abstract:** The recognition accuracy of traditional image recognition methods is heavily dependent on the design of complicated and tedious hand-crafted features. In view of the problems of poor accuracy and complicated feature extraction, this study presents a methodology for the estimation of the severity of wheat Fusarium head blight (FHB) with a small sample dataset based on transfer learning technology and convolutional neural networks (CNNs). Firstly, we utilized the potent feature learning and feature expression capabilities of CNNs to realize the automatic learning of FHB characteristics. Using transfer learning technology, VGG16, ResNet50, and MobileNetV1 models were pre-trained on the ImageNet. The knowledge was transferred to the estimation of FHB severity, and the fully connected (FC) layer of the models was modified. Secondly, acquiring the wheat images at the peak of the outbreak of FHB as the research object, after preprocessing for size filling on the wheat images, the image dataset was expanded with operations such as mirror flip, rotation transformation, and superimposed noise to improve the performance of the model and reduce the overfitting of models. Finally, under the Tensorflow deep learning framework, the VGG16, ResNet50, and MobileNetV1 models were subjected to transfer learning. The results showed that in the case of transfer learning and data augmentation, the ResNet50 model in Accuracy, Precision, Recall, and F1 score was better than the other two models, giving the highest accuracy of 98.42% and F1 score of 97.86%. The ResNet50 model had the highest recognition accuracy, providing technical support and reference for the accurate recognition of FHB.

**Keywords:** fusarium head blight; convolutional neural network; deep learning; diseases; transfer learning; ResNet50 model

## 1. Introduction

Wheat is one of the three most important food crops. In China, wheat is the second-largest food crop after rice [1]. It has evolved into the primary crop for human consumption and cattle feed because of its great yield potential, rich nutritional value, and excellent adaptability [2]. There is an economically devastating disease of wheat known as fusarium head blight (FHB), also known as scab, which is mainly caused by Fusarium asiatica and Fusarium graminearum [3]. White, shriveled wheat ears and a great reduction in wheat yield will result from the infestation of FHB [4]. In addition, wheat infected with FHB has poor quality, and its production is impacted. Moreover, FHB-infected crops have terrible grain quality, with dry, discolored grain, leading to significant losses and price drops [5]. What is worse, infected wheat will produce many mycotoxins, especially deoxynivalenol (DON) and zearalenone (ZEA), which are harmful to humans and animals. In addition to acute poisoning symptoms, this can also lead to impaired immunity and even death [6].

As a result of FHB, food security, high quality, and high yield as well as efficient and sustainable development are compromised. There is an urgent need to address the safety problems caused by FHB. In order to maintain agricultural production and prevent disease

spread, the severity of FHB must be accurately identified and pesticides must be applied rationally [7–9]. At present, the identification of FHB mainly relies on farmers' experience in many countries and areas, often by observing changes in infected areas on wheat ears [10]. It is challenging to quantitatively analyze and evaluate the occurrence of FHB in association with the actual condition of wheat due to the impact of human subjective factors. It is easy to overlook diseases and disasters since the diagnostic criteria are not standardized, there are issues with low efficiency, laboriousness, and strong subjectivity, and the authenticity and quality of the survey data cannot be guaranteed. Therefore, the fast, highly automated, and accurate diagnosis and identification of FHB is the foundation for precise pesticide application according to the severity of FHB and the affected wheat area, which helps to save pesticides, improves efficiency, lowers the cost, decreases dependence on laborers, and cuts down the pollution of pesticides to the agroecological environment, and plays an essential role in ensuring the high yield and high quality of wheat grain crops.

Machine learning technology has been created to boost efficiency. As computer processing power has expanded substantially, there has been significant development of machine learning technology coupled with image processing in the identification of crop diseases, with promising results being achieved [11,12]. The role of machine learning technology coupled with image processing in the diagnosis of crop diseases has received increased attention across several disciplines in recent years. There have been many efforts made by researchers to automate the identification of crop diseases such as those of rice, corn, soybeans, grapes, citrus, etc. using machine learning, image processing, and other technologies [13–21]. By manually extracting the color, shape, texture, and other features of the disease and using Fuzzy Logic (FL) classification [20], Artificial Neural Network (ANN) [14–17], AdaBoost [21], Decision Tree (DT) [21], Naive Bay (NB) [18], and Random Forest (RF) [19,21] crop disease classification was achieved. Although machine learning technology has made many achievements in plant disease identification, these studies all extract disease features through artificial design. When using classical machine learning methods for plant disease identification, it is necessary to extract plant disease features that have a great impact on the identification accuracy. Despite this, the use of classical machine learning techniques to determine the severity of crop disease is limited due to slight differences in color and texture of crops infected by different diseases. With classical machine learning methods, crop diseases cannot be accurately identified, and feature extraction and portability are problematic. Popularizing and utilizing it on a large scale is challenging.

In recent years, in addition to the above methods and models, researchers have shown an increased interest in deep learning. Deep learning methods, as a breakthrough in the field of computer vision, are also extensively utilized in image recognition and classification. Following the great success of the AlexNet [22] model in the ImageNet Large-Scale Visual Recognition Challenge (ILSVRC) competition in 2012, many convolutional neural network (CNN) models have been successively adopted because of their excellent performance promoted during the development of CNNs. It is proposed that general models include GoogleNet [23], VGGNet [24], ResNet [25], etc., and are used in the task of disease identification and classification of different crops. Agarwal et al. [26] proposed a simple CNN model method constructed by eight hidden layers for the identification of tomato disease, and the model achieved an identification accuracy of 98.7%. Liu et al. [27] constructed a Kiwi-ConvNet model for the identification of kiwifruit leaf disease and improved the feature extraction ability with the help of the idea of multi-dimensional feature fusion. The results of this study showed that the identification accuracy of this model could reach 98.54%. Fang et al. [28] used an adaptive adjustment algorithm to process images and optimized and improved the ResNet50 model to classify and identify the leaf disease levels of ten diseases in eight crops, and the recognition accuracy reached 95.61%. By using deep learning, we can circumvent the tedious task of manual feature extraction, and high accuracy can be achieved when there are enough training samples. At present, in the field of intelligent agriculture, deep learning methods have been widely used in the identification of diseases and insect pests or fine-grained crop disease identification.

Although CNN is effective in the identification and classification of crop diseases, its biggest restriction is that enough supervised learning samples are required in the model training stage to improve the capacity of image feature extraction [29]. However, in most practical situations, we can only obtain a limited number of training samples of the objects that need to be recognized [30–32]. In the case of a small number of learning training samples, the performance of the general CNN models is very poor. This is because of the lack of sufficient training samples, the network will have the problem of underfitting in the process of training and learning [33]. In addition, deep learning methods require a powerful graphics processing unit (GPU) to accelerate the training and learning process of the models, and complex CNN models often require more time to adjust parameters. Therefore, training a large CCN model or collecting a large amount of image data is an extremely time-consuming process. In this situation, a technique called transfer learning is widely used [34], which alleviates the problem of a large number of training samples needed for deep learning models. It is also widely used in the field of image identification. By using large datasets to train CNNs, transfer learning enables them to extract features more effectively. Pre-training networks have been trained using standard datasets, such as PlantVillage [35] and ImageNet [36], which can effectively reduce the number of samples required for network model training, and the model performance is also very excellent. Using the MobileNet backbone network and transfer learning, Chen et al. [37] implemented the identification and detection of rice diseases. It had been shown in many comparative studies that this method performed well, reaching an accuracy of approximately 98.48%. Shah et al. [34] used a pre-trained VGG16 model and performed a barley classification task based on transfer learning techniques, with a model accuracy rate of 94%. Aravind et al. [38] used six pre-trained CNN models based on transfer learning techniques to achieve the efficient classification of ten different diseases in four main crops.

It is generally believed that the above studies focus on identifying common leaf diseases and insect pests. So far, however, there has been little discussion about estimating the panicle diseases and the severity of the diseases. At present, only the categories of wheat diseases have been identified, which makes it challenging to apply pesticides precisely where they are needed. In addition, training a large CNN model is time-consuming and requires high hardware equipment. The interference of a complex environment will also affect the accuracy of the models. In response to the above problems, a measure has been taken to retain a single black background by placing a black baffle directly behind the wheat ears and to collect five severity images of FHB without the interference of the complex background in the field. After the preprocessing based on size filling, the dataset has been expanded through data augmentation operations such as image flipping, rotation transformation, superimposed noise, brightness transformation, etc. Finally, based on VGG16, ResNet50, and MobileNetV1, by modifying the last FC layer of the three models to adapt to the estimation of the severity of FHB and then using the transfer learning method to explore, a model with higher recognition accuracy is trained with smaller training samples in order to achieve accurate estimation of the severity of FHB.

## 2. Materials and Methods

### 2.1. Study Area and Experimental Design

The study area was located on an experimental farm, owned by the Xuchang Campus of Henan Agricultural University in Xuchang City, Henan Province, China, at approximately 34°8′ N, 113°47′ E. The terrain in this area was flat, and irrigation and drainage were also quite convenient. The soil type of the experimental farm was loamy, the previous crop was corn, and the fertility level was relatively high. The tested varieties were "Sumai 3", "Yangmai 158", "Ningmai 9", and "Zhoumai 18". Among them, 'Sumai 3' is a highly resistant variety, 'Yangmai 158' is a moderately susceptible variety, 'Ningmai 9' is a moderately resistant variety, and 'Zhoumai 18' is a highly susceptible variety.

From 2019 to 2020, the test varieties of wheat were used for autumn sowing. A total of 60 experimental plots were set up in the FHB inoculation experiment and were planted

in 3 rows, with 20 experimental plots in each row. Each plot was approximately 1 m by 1.5 m, and 6 rows of the same variety of wheat were sown in each experimental plot; the row spacing was 20 cm. The test varieties of wheat in the 3-row experimental plot were randomly planted, and management measures such as fertilization, weeding, and irrigation in the entire experimental plot were carried out. All were the same. In April 2021, during the flowering period of wheat, a single flower inoculation was used to inoculate the pathogen. The pathogen was the Fusarium graminearum strain. A total of 100 wheat ears were randomly selected from each plot as the target of inoculation. At the early stage of wheat flowering (10% of wheat ear blossoming in the middle), 20 μL of spore suspension was injected into a small flower in the middle and upper part of the wheat ear with a micropipette, and then the wheat ear was moistened by bagging for 1–7d. Then, the wheat ear was marked via awn shearing.

## 2.2. Severity of FHB

FHB is classified according to the proportion of the diseased spikelets with ear rot symptoms (or white ear symptoms caused by stalk rot) to all the spikelets. The specific grading standard in this study refers to the rules for monitoring and forecast of the wheat head blight (Fusarium graminearum Schw./Gibberella zeae (Schw.) Petch) (GB/T15796-2011). The grading standard of wheat FHB-diseased spikelets is divided into 5 levels according to the proportion of diseased spikelets to all spikelets: level 0, no disease; level 1, the number of diseased spikelets accounted for Less than 1/4 of all spikelets; level 2, diseased spikelets accounted for 1/4 to 1/2 of all spikelets; level 3, diseased spikelets accounted for 1/2 to 3/4 of all spikelets; level 4, the number of diseased spikelets accounted for more than 3/4 of all spikelets. Figure 1 lists the characteristics of the incidence of FHB of each severity.

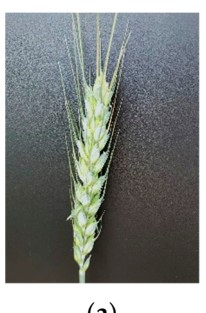 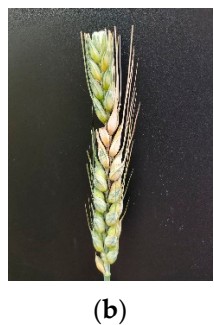 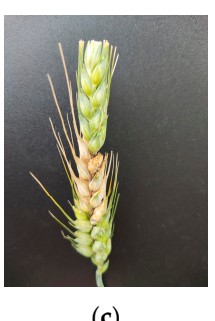 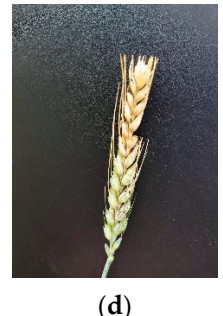 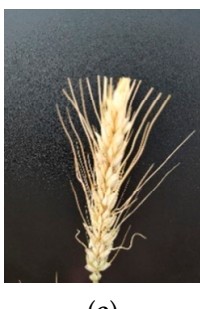

(**a**)　　　　　　(**b**)　　　　　　(**c**)　　　　　　(**d**)　　　　　　(**e**)

**Figure 1.** Comparison of various severity levels of FHB. (**a**) The typical image of FHB with a severity level of 0. (**b**) The typical image of wheat FHB with a severity level of 1. (**c**) The typical image of FHB with a severity level of 2. (**d**) The typical image of FHB with a severity level of 3. (**e**) The typical image of wheat FHB with a severity level of 4.

## 2.3. Data Acquisition and Enhancement

The single-ear sample data of FHB with five severity levels were captured at the experimental farm of the Xuchang Campus of Henan Agricultural University Xuchang Campus. In our experiment, black baffles were used to avoid the interference of the complicated background. May is the peak period of the outbreak of FHB. During the wheat filling period, image data of FHB were acquired on 7 May 2021, 13 May 2021, and 20 May 2021. Each time the image data acquisition was obtained in sunny weather, each shot was conducted on a vivo iQOO Neo3 mobile phone. A vivo iQOO Neo3 mobile phone has 48 million pixels in the rear cameras, and the aperture lens is f/1.8. Before shooting, a single wheat ear was placed in front of the blackboard, and the lens was approximately 20–30 cm away from the wheat ear. The resolution of the collected pictures varied in size, and a total of 238 pictures were taken.

In our work, in order to further augment the number of samples, lower the risk of overfitting to a certain extent, and improve the generalization capability of the models, we intended to elaborate on the collected wheat images by superimposing Gaussian noise, salt and pepper noise, mirror flip, rotation (90°, 180°, 270°), and brightness transformation operations to expand the image dataset. The expanded image dataset was 4760, of which there were 400 images of healthy wheat (FHB severity level 0) and 1840, 540, 540, and 1440 images of FHB severity level 1, FHB severity level 2, FHB severity level 3, and FHB severity level 4, respectively. The data enhancement and division are shown in Table 1.

**Table 1.** Distribution of original data, enhanced data, and divided data.

| Category | Number of Original Samples | Number of Samples after Data Enhancement | Number of Samples in the Training Set | Number of Samples in the Test Set |
|---|---|---|---|---|
| 0 | 20 | 400 | 320 | 80 |
| 1 | 92 | 1840 | 1472 | 368 |
| 2 | 27 | 540 | 432 | 108 |
| 3 | 27 | 540 | 432 | 108 |
| 4 | 72 | 1440 | 1152 | 288 |

### 2.4. Transfer Learning

Transfer learning [39] is based on pre-training models. Transfer learning is a machine learning method that utilizes existing knowledge to solve related research fields but different research tasks. Its goal is to complete the transfer of knowledge between similar fields. For CNNs, transfer learning is to successfully apply the knowledge obtained from training on specific data sets to new research fields to be solved. Transfer learning reduces the training data and computing power required for the construction of deep learning models, can solve the fact that small sample data sets tend to overfit in the complicated network structures [40], and effectively shortens the training time of models. The process of transfer learning is shown in Figure 2.

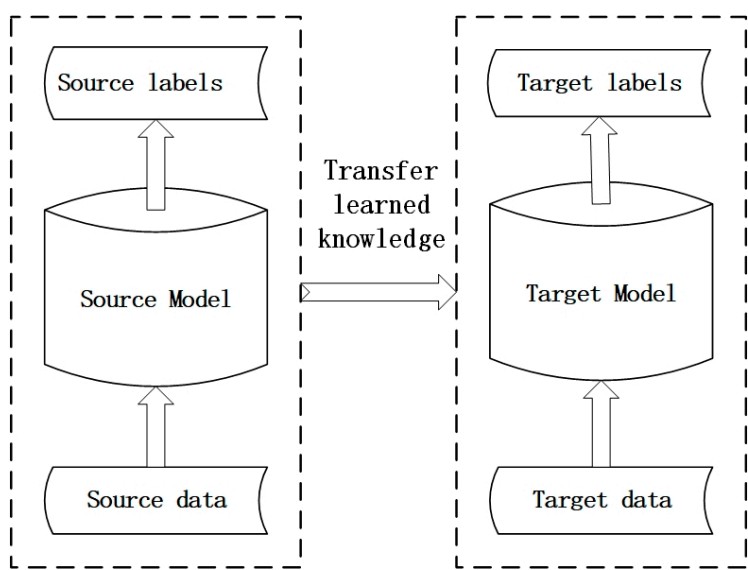

**Figure 2.** Transfer learning approach.

### 2.5. Model Construction

The CNN is a specialized feed-forward neural network with pipeline and multi-layer processing [41], and its network structure includes multiple convolutional layers, pooling layers, and FC layers. In CNN, the convolutional layer is the main component, which

extracts the features of the input images using different convolutional kernels. Use the following formula to calculate the convolution layer:

$$x^l_{j} = f(\sum_{i \in M_j} x^{l-1}_i k^l_{ij} + b^l_j) \tag{1}$$

where $x^l_j$ represents the output of the $j$th neuron at layer $l$, $x^{l-1}_i$ represents the output of the $i$th neuron at layer $l-1$, $M_j$ represents the input feature mapping set, $l$ represents the serial number of layers, $k^l_{ij}$ represents the convolution kernel, $b^l_j$ represents the bias, and $f(\cdot)$ represents the nonlinear activation function.

The nonlinear activation function often adopts the rectified linear unit (ReLu) function, and the expression of the ReLu function is:

$$f(x) = \begin{cases} 0 \ (x<0) \\ x \ (x \geq 0) \end{cases} \tag{2}$$

The convolutional layer is usually followed by a pooling layer. The pooling layer utilizes a pooling function to compress and reduce the dimension of the feature image and has translation invariance to the input, which can not only improve the model's ability to transform images such as displacement and rotation. It can also reduce the calculation amount and the number of parameters of the model. Commonly used pooling functions include average pooling, max pooling, and stochastic pooling. The process of pooling can be expressed by Equation (3):

$$x^l_j = f_{down}\left(x^{l-1}_i\right) \tag{3}$$

where $f_{down}(\cdot)$ is the down sampling function.

The FC layer is located after the alternation of multiple convolutional layers and pooling layers, and the extracted images are further reduced in dimension. Finally, the features are input into the softmax classifier for classification.

### 2.5.1. VGG16 Model

The VGG16 network [24] is one of the general CNN architectures proposed by the Visual Geometry Group of Oxford University, and is extensively used in image classification and object detection tasks. The VGG16 model achieved excellent outcomes in the 2014 ImageNet Image Classification and Localization Challenge, where it ranked second in the classification task and first in the localization task.

It consists of 13 convolution layers, 5 max-pooling layers, 3 FC layers, and 1 softmax layer. In this structure, all convolutional layers use the same $3 \times 3$ convolution kernels, and all convolutional layers and FC layers have ReLU nonlinear activation functions. The VGG16 network uses $3 \times 3$ convolution kernels, which reduces the number of parameters in the network and increases the nonlinear units in the network, which enhances the ability of the network to learn feature information and solve the problem of parameter explosion caused by large convolution kernels. The input image is passed through successively stacked convolutional layers and pooling layers to obtain the main feature information in the image and compress it. Finally, the learned image information is integrated and classified through the FC layer and the output layer. Its network structure is shown in Figure 3.

### 2.5.2. ResNet50 Model

The Residual Network proposed by He et al. [25] can alleviate the problems of vanishing gradient, exploding gradient, and network degradation to a certain extent due to the addition of residual units. The residual network adds skip connections between convolutional layers, so information can be spread across multiple hidden layers, which effectively alleviates the vanishing gradient and network degradation problems of CNNs so that the network depth can reach dozens or even hundreds of layers.

The residual unit is the vital structure of the residual network. The residual unit contains an identity map, which allows the input feature map to be transferred directly to the output without convolution. Even if the network depth is increased, the error remains constant. Figure 4 depicts the residual unit structure. When x is input into the network structure, there will be two lines. The middle one is obtained by the residual mapping of the weight layer and the ReLU function to obtain F(x), and the other one does not go through the operation of the weight layer; the identity mapping obtains x. The residual network learning feature obtained by adding the two lines is F(x) + x.

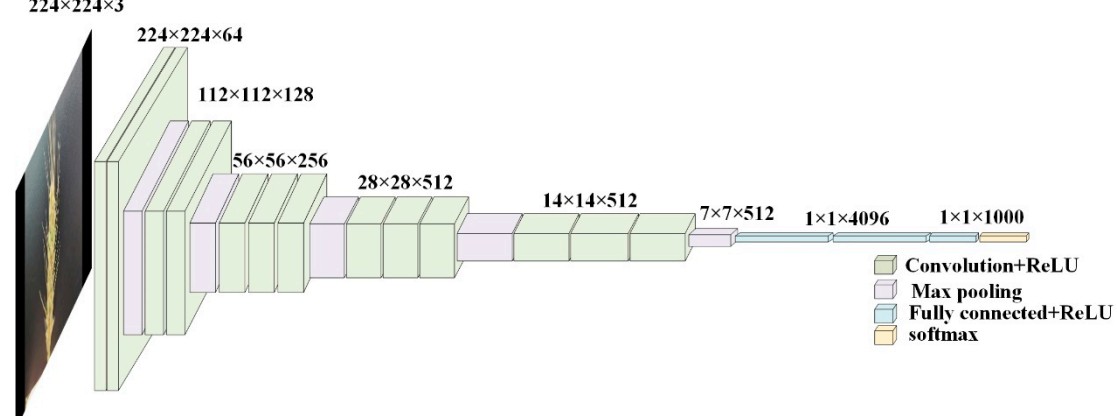

**Figure 3.** Structure of the VGG16 model.

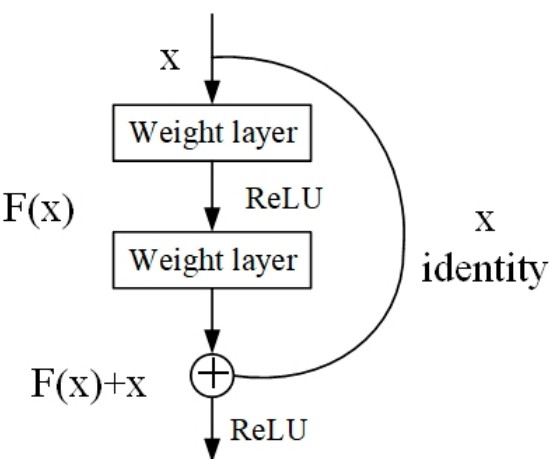

**Figure 4.** Structure of Residual unit.

2.5.3. MobileNetV1 Model

MobileNetV1 model is the first version of Mobilenet series networks, a network model proposed by Howard and colleagues [42]. MobileNetV1 model is a streamlined architecture from top to bottom, as shown in Figure 5. The MobileNetV1 model consists of a standard convolution layer (Conv Std), 13 depthwise convolution layers (Conv dw), 13 pointwise convolution layers (Conv pw), an average pool layer (Avg Pool), and an FC layer. The basic concept is depthwise separable convolution. It substitutes the common pool operation in convolutional neural networks by the depthwise convolution with a stride of 2, and only keeps the global average pool layer at the end of the network. In addition, there is a batch normalization layer (BN) and a ReLU after each convolution layer.

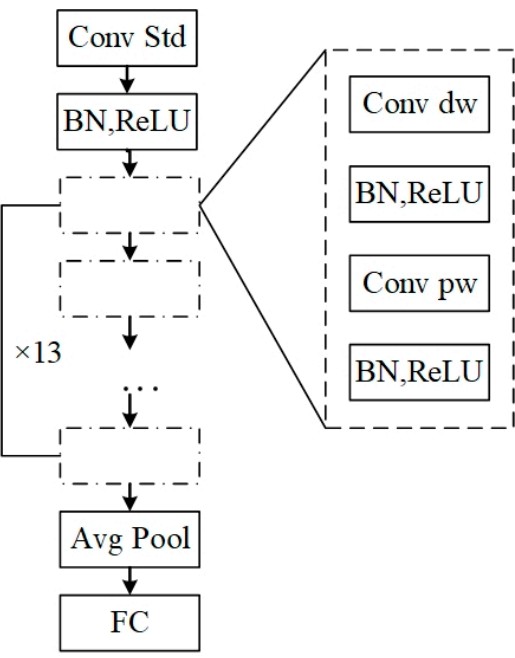

**Figure 5.** Structure of MobileNetV1 model.

*2.6. Model Optimization*

CNNs usually minimize the loss function to achieve the training goal. In this study, the stochastic gradient descent method (SGD) is used to optimize the model and cross-entropy is used as the loss function. The calculation formula is:

$$L = \frac{1}{N} \sum_i L_i = \frac{1}{N} \sum_i - \sum_{c=1}^{M} y_{ic} log(p_{ic}) \tag{4}$$

here, $M$ is the total number of categories; $y_{ic}$ is the indicating variable; if the true category of sample $l$ is $c$, take 1, otherwise, take 0; and $p_{ic}$ is the prediction probability that the observed sample $i$ belongs to category $c$.

In the process of training the models, the learning rate is also one of the significant parameters of deep learning. Choosing an appropriate learning rate can accelerate the convergence of the model and avoid the situation where the model oscillates near the minimum value. In this study, the cosine annealing [43] algorithm was adopted to update the learning rate in the process of training the models. The cosine annealing algorithm utilizes a cosine function and adopts the learning rate adjustment method that first slowly decreases, then accelerates, and finally decreases slowly. After the learning rate decays to 0 each time, it quickly returns to the initial value, and the learning rate is increased by a periodic restart. Accelerate the convergence speed of the model, reduce the learning rate, and slow down the convergence speed to jump out of the local minimum. Finally, find the path of the global minimum and approach the global optimal solution. In this study, the minimum value of the learning rate is set to 0.0001, and the calculation formula for the change of the learning rate with the number of epochs is shown in Equation (5):

$$\eta_t = \eta_{min}^i + \frac{1}{2}\left(\eta_{max}^i - \eta_{min}^i\right)\left(1 + cos(\frac{T_{cur}}{T_i}\pi)\right) \tag{5}$$

where $i$ represents the current $i$th restart, $\eta_{max}^i$ and $\eta_{min}^i$ indicates the range of learning rate, $T_{cur}$ represents the number of epochs currently executed, and $T_i$ represents the total number of epochs in the $i$th run.

## 2.7. Performance Measure Metrics

In our work, Accuracy, Precision, Recall, and F1 scores were used as performance measure metrics of the CNN models. The performance metrics are calculated in Formulas (6)–(9):

$$\text{Accuracy} = \frac{\text{TP} + \text{TN}}{\text{TP} + \text{TN} + \text{FP} + \text{FN}} \tag{6}$$

$$\text{Precision} = \frac{\text{TP}}{\text{TP} + \text{FP}} \tag{7}$$

$$\text{Recall} = \frac{\text{TP}}{\text{TP} + \text{FN}} \tag{8}$$

$$\text{F1} = \frac{2 * \text{Precision} * \text{Recall}}{\text{Precision} + \text{Recall}} \tag{9}$$

where true positive (TP) represents the number of correctly classified positive samples, true negative (TN) represents the number of correctly classified negative samples, false positive (FP) represents the number of misclassified positive samples, and false-negative (FN) represents the number of misclassified negative sample.

## 2.8. Model Training

### 2.8.1. Training Environment of Model

In our work, the operating system of the test platform is windows 10, 64-bit, and the training and testing of the models are conducted on Anaconda-Python 3.8 (Austin, TX, USA) and Tensorflow-GPU 2.2.0. The loading environment includes CUDA v10.1 and CUDNN v7.6. The CPU adopts AMD Ryzen 7, 4800H with Radeon Graphics with 8 CPU cores and 16 GB memory, and the CPU Clock Speed is 2.90 GHz. the GPU adopts NVIDIA GeForce RTX 2060 with 6 GB video memory (Shanghai, China).

### 2.8.2. Hyperparameter Design

In this study, we converted the input images of the three models with the fixed size of 224 × 224, and the three models pre-trained on the ImageNet dataset were utilized to modify the last FC layer of the three models. To adapt to the FHB dataset, the output was transformed into five different categories of outputs. Finally, the improved models were applied to the FHB severity dataset to train them. GPU was being used to accelerate the models' training and testing. Taking into account the performance of hardware equipment and the training time of models, in the process of model training and testing, the number of samples (Batch Size) of each epoch was set to 16, the initial learning rate was 0.01, and all three models executed for 70 epochs. The SGD optimization algorithm was used in model training to realize the optimization process of the model. The learning rate optimization strategy adopted the cosine annealing decay algorithm, the momentum was set to 0.9, the parameter settings of the three models were the same, and the training set and the test set were divided according to the ratio of 8:2, in which the training set included 3808 images and the test set included 952 images.

## 3. Results

### 3.1. Comparative Analysis of Results

In this work, we used the test set to conduct a comparative experiment of the classification performance of VGG16, ResNet50, and MobileNetV1 models for the assessment of the severity of FHB in order to evaluate and analyze the benefits and drawbacks of the three models. The results of the three models are shown in Figure 6 below. As can be seen from Figure 6, with the increase of epochs, the accuracy rates of the three models showed a gradually increasing trend and the losses of the three models showed a gradual decrease trend. Finally, the three models were all in 70 epochs and tended to converge. Among them, the ResNet50 model had the best performance, with an accuracy of 98.42%, and the model was converging fast, tending to converge around the 40th epoch, while the VGG16 model

had the slowest convergence, tending to converge around the 60th epoch. With certain fluctuations, the final accuracy reached 97.16%, which was slightly lower than the ResNet50 model, and its loss was slightly larger than that of the ResNet50 model. The MobileNetV1 model had the worst effect, with a classification accuracy of 92.75% and the largest loss.

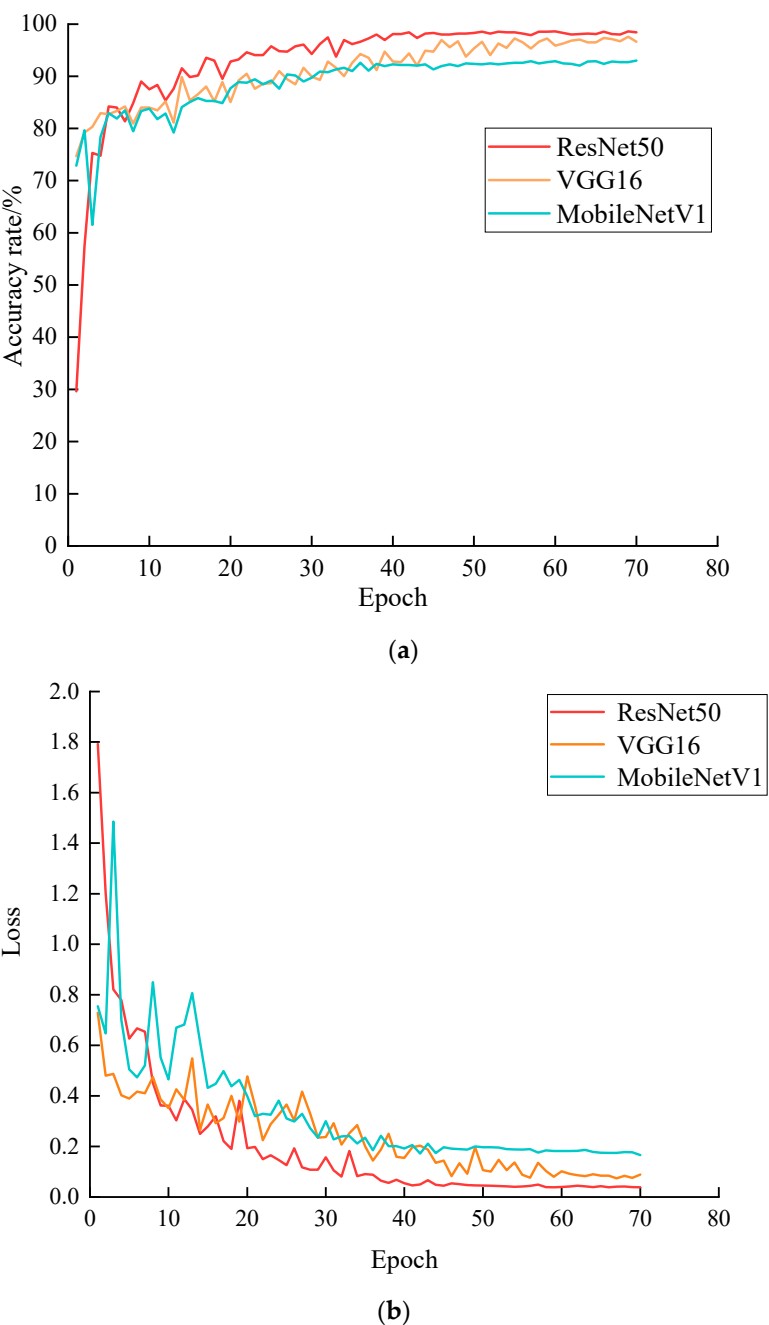

**Figure 6.** The accuracy (**a**) and the cross-entropy loss (**b**), respectively. (**a**) The accuracy of VGG16, ResNet50, and MobileNetV1 models changed with epochs. (**b**) The loss of VGG16, ResNet50 and MobileNetV1 models changed with epochs.

Table 2 illustrates the three models' Accuracy, Precision, Recall, and F1 scores. The accuracy of each model was better than 92%, as seen in Table 2, further demonstrating the significance of using CNN models to estimate the severity of FHB. Among these models, the ResNet50 model gave the best performance, with an accuracy of 98.42% in the test set and an F1 score of 96.03%, which was the most appropriate for estimating the severity of FHB.

**Table 2.** Performance Measure Metrics of the models.

| Models | Accuracy/% | Precision/% | Recall/% | F1/% |
|---|---|---|---|---|
| VGG16 | 97.16 | 96.37 | 95.69 | 96.03 |
| ResNet50 | 98.42 | 98.38 | 97.35 | 97.86 |
| MobileNetV1 | 92.75 | 91.30 | 87.57 | 89.40 |

*3.2. Confusion Matrix*

Figure 7 shows the confusion matrix of VGG16, ResNet50, and MobileNetV1 models. From the confusion matrix of the three models, it can be seen that the images of healthy wheat (severity level 0) and images of wheat that were almost completely infected with FHB (severity level 4) were rarely misclassified. This was especially true for VGG16 and ResNet50, two models with better performance, which were only slightly misclassified in the MobileNetV1 model. In addition, the misclassified samples of the three models were mainly concentrated on wheat images with severity level 2 and severity level 3, while the ResNet50 model had a better performance for severity level 2 and severity level 3. In the case of several samples being misclassified, the recognition effect of the VGG16 model was slightly inferior to that of the ResNet50 model, while the MobileNetV1 model could not distinguish FHB images of various severity well, and the misclassification phenomenon was more serious. In general, the ResNet50 model had the best effect, with the fewest misclassified samples.

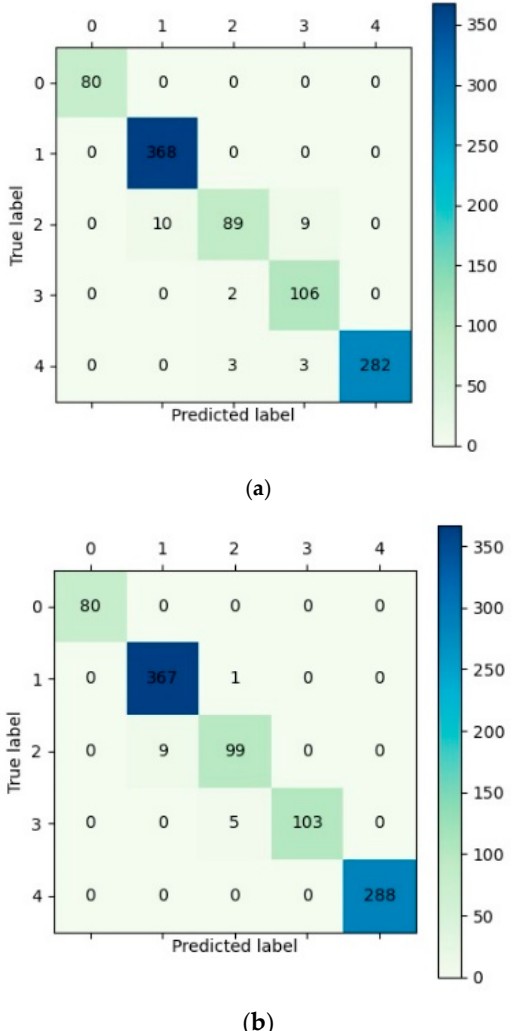

(**a**)

(**b**)

**Figure 7.** *Cont.*

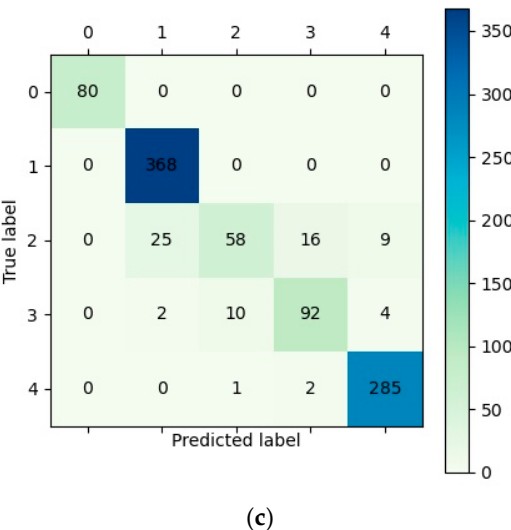

(**c**)

**Figure 7.** Confusion matrix of VGG16 (**a**), ResNet50 (**b**), and MobileNetV1 (**c**) models. (**a**) Confusion matrix of VGG16 model. (**b**) Confusion matrix of ResNet50 model. (**c**) Confusion matrix of MobileNetV1 model.

## 4. Discussion

Several reports have shown that traditional machine learning methods have been extensively used in crop disease identification, and the amount of work generally focuses on extracting color features, texture features, and shape features of the crop diseases to be studied. Because disease images obtained in natural environments are susceptible to natural illumination and occlusion, in order to lower the influence of brightness on disease feature detection, the RGB color space is often converted to color spaces such as HSV [17], HIS, Lab [17], and YCbCr [16], and the color features usually include color Mean, Entropy, Variance, Kurtosis, Skewness, etc. The most commonly used texture features are Contrast, Correlation, Energy, and other features based on Gray-level co-occurrence matrix (GLCM) [13]. The shape features mainly include Rectangularity, Eccentricity, Density, etc. In addition, in order to effectively segment crop disease spots, some segmentation algorithms, such as K-means [17], Roberts [44] Sobel [45], Graph Cut [46], etc., are also used.

With the development of CNN, many researchers have also used deep learning technology to realize plant identification [47], disease detection [48,49], weed detection [50] pest recognition [31,51], etc., and achieved better performance than traditional machine learning. Literature [26] proposed a CNN model for tomato disease identification and the accuracy was 98.7%, which was better than traditional machine models such as KNN, DT, and SVM. In apple disease identification research, Liu et al. [52] identified four apple diseases, including Mosaic, Rust, Brown spot, and Alternaria leaf spot, based on the improved AlexNet, with an accuracy of 97.62%, which was higher than the traditional BPNN and SVM model. However, training a powerful CNN often takes a lot of time and requires relatively powerful computing and memory resources. Deep learning models usually need to be completed on GPU devices, and deep learning models also require the use of a lot of training samples to prevent the model from underfitting. In addition, due to the excessive number of layers and the large number of parameters, the model may also be overfitted. The use of transfer learning technology can solve the problem of overfitting on complex models to a certain extent, and research the basis of learning in similar research fields, which relatively saves time.

Based on the powerful feature extraction function of CNN, for this research, we used the VGG16, ResNet50, and MobileNetV1 models pre-trained on the ImageNet large data set. We estimated FHB severity by modifying the FC layer structure to adapt to the image data set and integrating transfer learning techniques. The parameter settings of the three models were all the same. From the estimation results of the three models, they all reached

the convergence state before 70 epochs. Finally, the accuracy rates of the three models were all above 92%, which indicated that the CNN model based on transfer learning effectively realized the task of estimating the severity of FHB. Although the ResNet50 model had a deep network structure, it had the best performance for the task of estimating the severity of FHB, with an accuracy of 98.42%. Compared with the VGG16 model, the complexity of the ResNet50 model and the number of parameters required had decreased, and because of the addition of residual units, the network layers were deeper; however, the gradient did not disappear. Therefore, the overall performance was better. The effect of the VGG16 model was slightly inferior to the ResNet50 model, and its loss and accuracy had certain volatility during the training process, which indicated that the model converged slowly during the training process and was inferior to the ResNet50 model and the MobileNetV1 model in the stability of model training. The result of MobileNetV1 only reached 92.75%. The reason for this unsatisfactory outcome may be that the model is lightweight, and the reduction in the number of parameters makes it difficult for the model to explore abundant information for the estimation of FHB severity, which does not achieve the desired result.

From the confusion matrix of the three models, it can be seen that the misclassified samples of models were mainly concentrated in the samples with severity level 2 and severity level 3. The possible reason was that the number of samples of these two severities was relatively small. The incidence areas between the two severities in the collected samples were relatively similar, and the difference may not have been obvious enough, resulting in a relatively large number of misclassified samples. The results of the VGG16, ResNet50, and MobileNetV1 models showed that it was feasible to use transfer learning to realize the severity estimation task of FHB, and could achieve good results, which could provide a reference for the disease severity diagnosis and precise drug application of FHB.

In addition to the methods used above, in this study, we also used traditional methods to realize the severity estimation of FHB. Because the color of FHB will be yellow and dry compared with the normal ears after the infection of FHB, we can make full use of color features to realize the severity estimation of FHB. In order to compare and analyze with the method using CNN, the image of the test set was used to carry out the research, and the size of the image was also redefined as 224 × 224. This study used the Otsu threshold segmentation method to segment the affected area of FHB. Firstly, in order to reduce the impact of natural light, the obtained RGB image was converted to HSV color space and Lab color space. Secondly, three components of RGB color space, three components of HSV color space, and three components of Lab color space were extracted. Through comparison, it was found that the component of Lab color space could clearly identify the infected area. The comparison of Otsu segmentation results of different components in three color spaces can be clearly seen in Figure 8. Therefore, the subsequent experiments were carried out using a component combined with threshold segmentation. Finally, the recognition accuracy was 64.18%, and the accuracy did not reach satisfactory results. After analysis, it was found that the reason may be that the image acquisition was carried out when the light was good, the brightness of some images was high, and the threshold segmentation of the image was affected to a certain extent, resulting in the background part being unable to be well-separated from the wheat ear part. Another reason may be that the classification task of this study was divided according to the proportion of the area infected with FHB in the total area of wheat ears. Subtle differences in the segmentation process may lead to large differences in the classification results of the image. From the results, it can be seen that compared with the traditional methods, the CNN used in this study has unparalleled advantages and can provide very accurate information on the severity of FHB. Therefore, it is very meaningful to use CNN and transfer learning methods to realize the severity estimation of FHB.

In future work, we will expand the dataset by adding more varieties of wheat images and further enrich the data with data augmentation algorithms to improve the generalization ability of the model. Improving the imbalance of samples in different categories of

data is also one of the future research works. In addition, in terms of the algorithm used, the improvement of the estimation accuracy of the model is the top priority of future work.

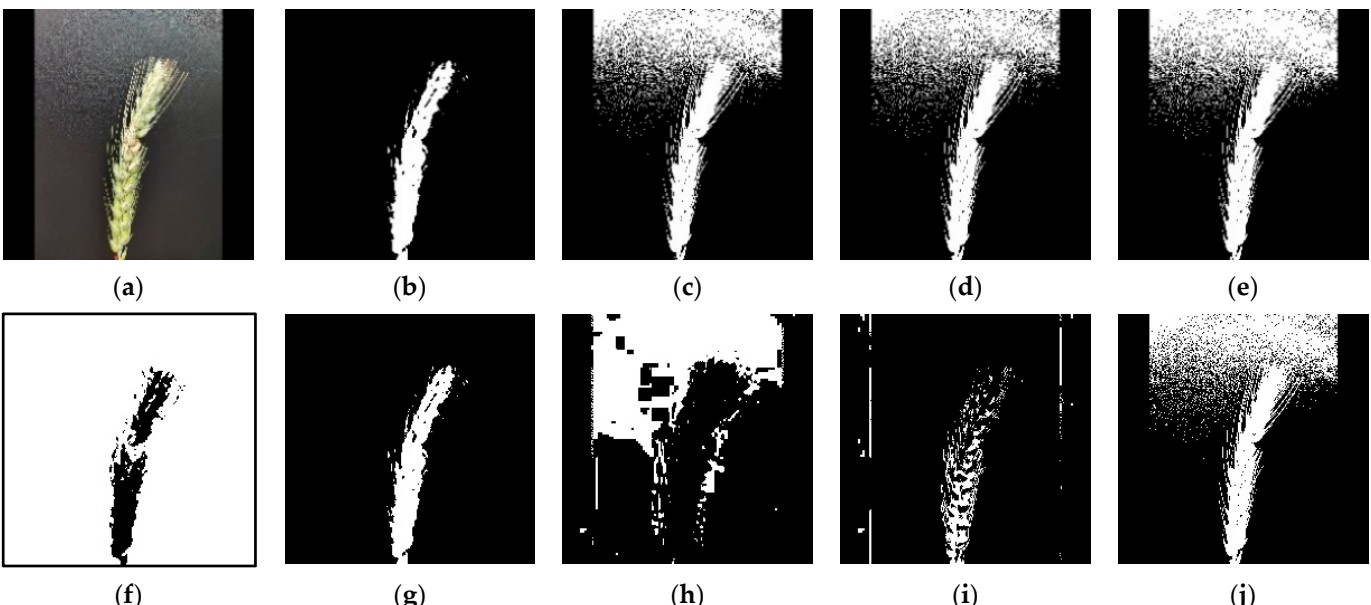

**Figure 8.** Otsu segmentation results of different components in three color spaces. (**a**) Original image. (**b**) RGB-B component. (**c**) RGB-G component. (**d**) RGB-R component. (**e**) Lab-L component. (**f**) Lab-a component. (**g**) Lab-b component. (**h**) HSV-H component. (**i**) HSV-S component. (**j**) HSV-V component.

## 5. Conclusions

FHB is one of the main diseases affecting the yield and quality of wheat. Therefore, timely identification and diagnosis of the severity of FHB is the premise of accurate application. In view of the problem that traditional machine learning methods need to manually extract features, this study used CNN models to estimate the severity of FHB. In addition, considering the training time and hardware equipment of the models, based on the VGG16, Resnet50, and MobileNetV1 model structures, this study modified the FC layer structure to estimate the five severities of FHB. Firstly, the pre-training models obtained in ImageNet were used for transfer learning, and then the data set was expanded by combining the image enhancement strategy. The test set accuracy of the ResNet50 model was 98.42%, which was better than the other VGG16 model and MobileNetV1 model. The results showed that the ResNet50 model was more suitable for estimating the severity of FHB.

**Author Contributions:** Conceptualization, C.G. and Z.G.; methodology, C.G.; software, Z.G.; validation, C.G., Z.G. and W.G.; investigation, M.D., Q.H. and H.S.; resources, W.G. and C.G.; data curation, W.G.; writing—original draft preparation, C.G. and W.G.; writing—review and editing, W.G. and C.G.; visualization, Z.G. and X.J.; supervision, W.G. and C.G.; funding acquisition, W.G. and X.J. All authors have read and agreed to the published version of the manuscript.

**Funding:** This research was funded by the Henan Province Science and Technology Research Project (212102110028, 22102320035); the National engineering research center for Argo-ecological big data analysis and application (AE202005); and the Science and technology innovation fund of Henan Agricultural University (KJCX2021A16).

**Institutional Review Board Statement:** Not applicable.

**Informed Consent Statement:** Not applicable.

**Data Availability Statement:** Not applicable.

**Conflicts of Interest:** The authors declare no conflict of interest.

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
