# Peer review of "Estimation of Fusarium Head Blight Severity Based on Transfer Learning"

_agronomy, doi:10.3390/agronomy12081876_

Round 1

Author Response

Dear reviewer:

I would like to express my sincere thanks to you for your careful review of this article and your valuable comments. I have modified and improved the problems in the article one by one according to your suggestions and requirements, in order to modify the integrity of the content, the rigor of logic and the rationality of the structure. The following are the replies to your modification suggestions one by one.

Point 1: page 4 (row 151-152) the wheat cultivars tested in the study are listed. “The tested varieties were “Sumai 3”, “Yangmai 158”, “Ningmai 9”, and “Zhoumai 18”. Unfortunately there is no further information about the cultivars. The reader does not know how much they differ in susceptibility to FHB. It would be also good to inform whether cultivars triggered any differences in the results of the experiment. It is particularly interesting whether there were any differences between the cultivars without Fusarium head blight symptoms. A short comment about this needs to be added.

Response 1: Dear reviewer, this problem has been further explained in the manuscript about the susceptibility of wheat varieties. Among them, 'Sumai 3' is a highly resistant variety, 'Yangmai 158' is a moderately susceptible variety, 'Ningmai 9' is a moderately resistant variety and 'Zhoumai 18' is a highly susceptible variety (page 4, row 152-154). Wheat varieties without Fusarium head blight symptoms only show differences in grain size and grain number in appearance, and there are also differences in disease resistance and the time difference of the key growth period of wheat. The different test varieties make this study applicable to more wheat varieties, so as to be extended to greater agricultural applications.

Point 2: At page 4 (row 159) reader gets the following information: “In April 2021, during the flowering period of wheat, a single flower inoculation was used to inoculate pathogenic bacteria.” While Fusarium belongs to the Kingdom of fungi.

Response 2: Dear reviewer, thank you for raising this question. I'm sorry for the word mistake. I have carefully corrected this problem and added pathogen information (page 4, row 161-163).

Point 3: At page 5 (row 193) the reader gets the following information: “a total of 238 pictures were taken”. While in table 1. Number of original samples is: 19, 92, 27, 27, 72 which in total gives 237.

Response 3: Dear reviewer, thank you for your careful review. I carefully verified that the actual number of pictures is 238, and the number of 19 in the first row of Table 1 in the manuscript has been correctly changed to 20 (page 5, row 206).

Point 4: Moreover the presented pictures differ in the background. Actually the background of the picture presented the ear with no symptoms differs with the backgrounds of the others. It needs to be corrected.

Response 4: Dear reviewer, thank you very much for your reminder. I have changed it to display image information according to your requirements (page 4, row 180).

Reviewer 2 Report

This paper proposed a methodology for estimating FHB based on the transfer learning technology and convolution neural network. The results show that the accuracy, precision, recall, and F1 score are above 95%. 

However, I have some questions as follows.

1. The authors claimed that one of the contributions is using transfer learning. This is a little bit weak since using the imagenet for pre-training is a standard process. And the authors did not show the model performance without pre-training on the imagenet. It will make more sense if the authors could use the images of other crop plants.

2. For Figure 6 (a), could the authors explain why the accuracy of VGG16 and MobileNet is much higher than that of ResNet50?

3. What's the growing stage of the wheat in the picture? Is the model able to detect the FHB in the early stage?

4. The images used for training are too ideal (high resolution, clear background). Have the authors thought about the application of this model to agricultural production? Maybe a farmer or a trained person is more efficient than taking a picture and uploading it to the model.

5.  This task is not challenging since FHB degree is largely determined by the ratio of the yellow part to the green part. The authors need to compare the CNNs with other traditional methods (e.g., filtering by color) and explain why machine learning is necessary here.

Author Response

Dear reviewer:

I would like to express my sincere thanks to you for your careful review of this article and your valuable comments. I have modified and improved the problems in the article one by one according to your suggestions and requirements, in order to modify the integrity of the content, the rigor of logic and the rationality of the structure. The following are the replies to your modification suggestions one by one.

Point 1: The authors claimed that one of the contributions is using transfer learning. This is a little bit weak since using the imagenet for pre-training is a standard process. And the authors did not show the model performance without pre-training on the imagenet. It will make more sense if the authors could use the images of other crop plants.

Response 1: Dear reviewer, thank you very much for your opinion. First of all, considering the number of data sets used in this study, in addition, considering the training time and efficiency, this study finally uses the standard data set ImageNet for the pre-training process. Due to the limitation of the collected data, it is regrettable that we have not obtained the image data set of the same type of crop plants at present. If possible, we will conduct a deeper study later.

Point 2: For Figure 6(a), could the authors explain why the accuracy of VGG16 and MobileNet is much higher than that of ResNet50?

Response 2: Dear reviewers, I am very glad to answer your questions. The result of this study was that the accuracy of ResNet50 and VGG16 is higher than that of MobileNet. The reason may be that the ResNet50 model has a deep network structure. Compared with the VGG16 model, the complexity and the number of required parameters of the ResNet50 model have decreased, and due to the increase of residual units, the network layer is deeper, but the gradient does not disappear. Therefore, the overall effect is the best. The effect of the VGG16 model is slightly inferior to the ResNet50 model, and its loss and accuracy fluctuate in the training process, indicating that the model converges slowly in the training process, and the accuracy of the MobileNet model only reaches 92.75%. The reason may be that the model is lightweight, and the reduction of parameters makes it difficult for the model to explore rich information to estimate the severity of FHB, resulting in the accuracy being lower than the other two models. The further analysis of the results is reflected in the” Discussion “of Part 4 (page 13, row 424-452) in the manuscript.

Point 3: What's the growing stage of the wheat in the picture? Is the model able to detect the FHB in the early stage?

Response 3: Dear reviewer, the wheat growth period at that time belongs to the filling period, which is also reflected in Section 2.3 (page 4, row 189-190). Wheat begins to be infected after inoculation at the flowering stage. In the early stage of wheat, as long as wheat is infected and has certain symptoms, the models used in this study can also detect FHB.

Point 4: The images used for training are too ideal (high resolution, clear background). Have the authors thought about the application of this model to agricultural production? Maybe a farmer or a trained person is more efficient than taking a picture and uploading it to the model.

Response 4: Dear reviewer, thank you very much for raising this question. The data set used in this study is indeed ideal. Considering the experimental results of this study, in future work, I will further carry out the severity estimation experiment of FHB, and collect images under natural conditions for further experimental research. Our future research work will fully consider these problems, Making it more suitable for application in agricultural production.

Point 5: This task is not challenging since FHB degree is largely determined by the ratio of the yellow part to the green part. The authors need to compare the CNNs with other traditional methods (e.g., filtering by color) and explain why machine learning is necessary here.

Response 5: Dear reviewer, according to your suggestion, we have supplemented the traditional method to realize the task of estimating the severity of FHB. In this study, the Otsu threshold segmentation method combined with HSV color space and Lab color space was used to realize the estimation of the severity of FHB. In addition, this part further explains the importance of machine learning. The improved content was marked in the manuscript (page 13-14, row 453-479).

Round 2

Reviewer 2 Report

The authors have answered my question. I agree to publication